# Targeting Acute Myeloid Leukemia with Venetoclax; Biomarkers for Sensitivity and Rationale for Venetoclax-Based Combination Therapies

**DOI:** 10.3390/cancers14143456

**Published:** 2022-07-15

**Authors:** Mila S. Griffioen, David C. de Leeuw, Jeroen J. W. M. Janssen, Linda Smit

**Affiliations:** Department of Hematology, Amsterdam UMC, Location VUmc, Cancer Center Amsterdam, 1081 HV Amsterdam, The Netherlands; m.s.griffioen@amsterdamumc.nl (M.S.G.); d.deleeuw@amsterdamumc.nl (D.C.d.L.); j.janssen@amsterdamumc.nl (J.J.W.M.J.)

**Keywords:** venetoclax (BCL-2 inhibitor), AML, biomarkers, resistance, sensitivity, therapeutic combinations

## Abstract

**Simple Summary:**

Venetoclax has proven to be a promising therapy for newly diagnosed, relapsed and refractory AML patients ineligible for induction chemotherapy. Current ongoing clinical trials are evaluating its effectivity as frontline therapy for all acute myeloid leukemia (AML) patients. However, response rates vary wildly, depending on patient characteristics and mutational profiles. This review elaborates on the efficacy and safety of venetoclax compared to conventional chemotherapy for treatment of AML patients, comparing the response rates, overall survival and adverse events. Moreover, it gives an overview of genetic and epigenetic AML cell characteristics that give enhanced or decreased response to venetoclax and offers insights into the pathogenesis of venetoclax sensitivity and resistance. Additionally, it suggests possible treatment combinations predicted to be successful based on identified mechanisms influencing venetoclax sensitivity of AML cells.

**Abstract:**

Venetoclax is a BCL-2 inhibitor that effectively improves clinical outcomes in newly diagnosed, relapsed and refractory acute myeloid leukemia (AML) patients, with complete response rates (with and without complete blood count recovery) ranging between 34–90% and 21–33%, respectively. Here, we aim to give an overview of the efficacy of venetoclax-based therapy for AML patients, as compared to standard chemotherapy, and on factors and mechanisms involved in venetoclax sensitivity and resistance in AML (stem) cells, with the aim to obtain a perspective of response biomarkers and combination therapies that could enhance the sensitivity of AML cells to venetoclax. The presence of molecular aberrancies can predict responses to venetoclax, with a higher response in NPM1-, IDH1/2-, TET2- and relapsed or refractory RUNX1-mutated AML. Decreased sensitivity to venetoclax was observed in patients harboring FLT3-ITD, TP53, K/NRAS or PTPN11 mutations. Moreover, resistance to venetoclax was observed in AML with a monocytic phenotype and patients pre-treated with hypomethylating agents. Resistance to venetoclax can arise due to mutations in BCL-2 or pro-apoptotic proteins, an increased dependency on MCL-1, and usage of additional/alternative sources for energy metabolism, such as glycolysis and fatty acid metabolism. Clinical studies are testing combination therapies that may circumvent resistance, including venetoclax combined with FLT3- and MCL-1 inhibitors, to enhance venetoclax-induced cell death. Other treatments that can potentially synergize with venetoclax, including MEK1/2 and mitochondrial complex inhibitors, need to be evaluated in a clinical setting.

## 1. Introduction

Acute myeloid leukemia (AML) is a rapidly progressing heterogeneous disease caused by the accumulation of malignant myeloid hematopoietic precursor cells in the bone marrow and, in some cases, in the peripheral blood. AML often affects the elderly, with a median age at diagnosis of 68 years [1]. The prognosis and response to treatment of AML patients depends on intrinsic patient characteristics, cytogenetic and molecular aberrancies present in the leukemic cells, and the epigenetic, metabolic and transcriptomic state of the cells [2]. AML has historically been classified based on the level of maturation of the AML cells following the French, American, British (FAB) system. The most differentiated AML cases are categorized as FAB-M5 while the FAB-M0 class constitutes the most immature cases with a stem cell-like state [3], but this classification proved to have little clinical value. More recently, the European Leukemia Net (ELN) classified AML based on specific prognostic cytogenetic and molecular features that are present in the leukemic cells, including mutations in NPM1, FLT3 and TP53 into three risk categories for relapse (favorable, intermediate or adverse risk), which has direct implications for therapeutic decisions [4].

Currently, standard treatment of newly diagnosed AML patients consists of intensive induction chemotherapy with cytarabine and an anthracycline, more commonly known as the “7 + 3” regimen [4], followed by consolidation therapy, which can consist of 2–4 cycles with intermediate dose cytarabine or an autologous or allogenic hematopoietic stem cell transplantation, depending on age and genetic characteristics [4]. When treated, 60–80% of younger patients achieve complete remission (CR) with a median overall survival (OS) of 16–24 months, and 40–60% of patients aged above 60 years achieve CR with a median survival of 9 to 12 months [4,5,6,7,8]. However, newly diagnosed elderly patients or those with comorbidities are often only offered supportive care or other palliative treatments due to poor tolerance for intensive chemotherapy in these patient categories [4]. Consequently, long-term cure rates for older AML patients have been as low as 5–20%, with the worst outcomes in high-risk subgroups [9]. The poor treatment outcome of AML is thought to be due to survival of small subpopulations of therapy-resistant leukemic cells known as measurable residual disease (MRD) [10,11]. Within MRD, leukemic cells with self-renewal capacity and stem cell features, referred to as leukemic stem cells (LSCs), are responsible for the initiation of relapse [12]. LSCs reside within the bone marrow together with normal hematopoietic stem cells (HSCs), which are crucial for regeneration of healthy blood cells after therapy. 

For patients considered ineligible for induction chemotherapy, treatment with hypomethylating agents (HMA), such as azacitidine or decitabine, or low-dose cytarabine, (LDAC) has recently been the standard of care [4]. LDAC monotherapy achieved CR rates of 15–25% with a median OS of only 5 to 6 months [13]. HMA monotherapy was demonstrated to modestly improve median OS to 10.4 months [14]. 

In 2018, venetoclax received accelerated approval by the FDA for use in AML patients. As a BH3 mimetic, venetoclax is a selective and oral inhibitor that binds to BCL-2 [15], an important regulator of the apoptosis pathway by tightly controlling the release of cytochrome C from the mitochondria, the initiating step during apoptosis [16]. More specifically, upon binding to BCL-2, venetoclax allows for the interaction of the pro-apoptotic proteins BAX and BIM, which together increase the permeability of the outer mitochondrial membrane (Figure 1A) [17,18]. Once cytochrome C is released from the mitochondria, the caspase cascade, or apoptosis pathway, is activated. Resisting cell death and preventing apoptosis is one of the hallmarks of cancer [19], and targeting therapy resistance through BCL-2 inhibition may be a potential successful therapeutic approach to induce apoptosis in AML cells. Overexpression of BCL-2 has been implicated in survival of AML cells and has been associated with resistance to chemotherapeutics and poor outcome in AML patients [20,21].

AML relapse is thought to be initiated by LSCs that have self-renewal abilities but may be cell cycle quiescent [22,23,24]. These quiescent cells require less energy and therefore have lower oxidative metabolism [25]. In contrast to HSCs, quiescent LSCs overexpress BCL-2 [25,26], making them more susceptible to venetoclax treatment. In leukemic cells, overexpression of BCL-2 is observed in AML cases belonging to the more immature FAB-M0/M1 subtypes [27]. BCL-2 can increase oxygen consumption and mitochondrial oxidative phosphorylation (OXPHOS), the primary energy source for proliferation and survival of cells [28]. LSCs have an increased dependency on oxidative metabolism compared to normal hematopoietic cells because of reduced glycolytic capacities [25]. Treatment of AML cells with the combination of venetoclax and azacitidine resulted in rapid eradication of LSCs, while sparing normal healthy cells [25]. The specificity for LSCs is due to a venetoclax-induced decrease in OXPHOS [25,29], caused by lower expression of transporters and therefore decreased amino acid uptake in the LSCs (Figure 1A) [30]. Treatment with venetoclax-based therapy specifically eradicates LSCs, as normal HSCs will compensate for reduced OXPHOS through increased glycolysis and will therefore survive [25].

Since its approval as a treatment for chronic lymphocytic leukemia (CLL), venetoclax treatment has primarily been investigated in AML patients unfit for or refractory to standard treatment regimens. Elderly AML patients, unfit for standard treatment with induction chemotherapy, showed a 67% CR/CR with incomplete blood count recovery (CRi) rate with a median OS of 17.5 months when treated with venetoclax and a HMA [31]. For comparison, AML patients above 60 years receiving induction chemotherapy achieved CR/CRi rates of 40–60% [6], suggesting that venetoclax-based therapy may achieve comparable responses as standard induction chemotherapy for the elderly AML patients. Therefore, the question to whether venetoclax–HMA combinations could be used as front-line treatment instead of induction chemotherapy for all AML patients needs to be addressed urgently.

Unfortunately, upfront resistance to venetoclax as well as relapse following CR still occurs [32]. The mechanisms that cause persistence of AML cells after venetoclax treatment include a variety of epigenetic, transcriptional and metabolic processes that often co-exist [33,34,35]. For the development of successful venetoclax-based combination therapy that efficiently eradicates MRD, and for identification of patients who will benefit from venetoclax treatment, it will be crucial to understand the mechanisms driving sensitivity and resistance to venetoclax in AML. This review elaborates on the efficacy and safety of venetoclax compared to conventional chemotherapy for the treatment of AML patients, comparing the response rates, overall survival and adverse events. Moreover, it gives an overview of genetic and epigenetic AML cell characteristics that have an enhanced or decreased response to venetoclax, and it suggests possible treatment combinations predicted to be successful based on the identified mechanisms that modulate sensitivity to venetoclax. 

## 2. Efficacy of Venetoclax as Compared to Chemotherapy in AML Patients

Most clinical studies of venetoclax-based therapy for AML patients were performed in patients aged > 60 years or unfit for conventional treatment with induction chemotherapy. In these clinical trials, venetoclax was given together with HMA, azacitidine or decitabine, or LDAC. The AML patients included had different stages of disease, some newly diagnosed, while others had relapsed or refractory disease. 

In newly diagnosed elderly AML patients treated with venetoclax-based therapy, CR/CRi was reached in 34–90% with a median OS ranging from 8.4 to 17.5 months and progression-free survival (PFS) ranging from 4.7 to 22.4 months (Table 1) [31,36,37,38,39,40,41,42,43,44]. In a meta-analysis comparing outcomes presented in 12 studies treating AML patients with venetoclax combination therapy (CR: 68%) [45], patients who achieved CR/CRi after venetoclax-based therapy had longer median OS (17 to 18.4 months) [39,41] than the overall cohort (8.4 to 17.5 months) (Table 1) [31,36,37,38,39,40,41,42,43,44]. The efficacy of venetoclax-based therapy appears superior to that of standard chemotherapy in patients > 60 years (induction chemotherapy: CR/CRi rate of 40–60%) [5,6]. The addition of venetoclax to LDAC or azacitidine significantly improved response rates compared to placebo [38,42]. In a propensity score-matched cohort, AML patients reached an MRD-negative composite CR rate of 86% when treated with venetoclax and intensive chemotherapy, compared to 61% when treated with only intensive chemotherapy [46]. Together, these data from clinical trials suggest that venetoclax-based therapy could improve response rates and survival for newly diagnosed elderly AML patients compared to standard chemotherapy. Randomized clinical trials comparing the efficacy of venetoclax-based treatment as front-line therapy compared to intensive chemotherapy are ongoing and are needed to confirm these initial findings (NCT04801797, NCT05177731, NCT05048615).

The efficacy of venetoclax was also analyzed in relapsed or refractory AML after standard chemotherapy, HMAs or a HSC transplantation [49]. Although less successful than the treatment of newly diagnosed AML, a meta-analysis showed that treatment with venetoclax-based therapy in these relapsed or refractory AML patients resulted in CR/CRi rates of 21% (venetoclax monotherapy) and 33% (venetoclax + HMA/LDAC) [55]. Median OS was 3.0 to 7.8 months (Table 1) [36,39,44,47,48,49,50,51,52,53,54]. In the patients who reached CR/CRi, PFS lasted 2.3 to 8.9 months, and median OS was 10.8 to 16.6 months (Table 1) [36,39,44,47,48,49,50,51,52,53,54]. Venetoclax monotherapy showed inferior results compared to venetoclax in combination with HMA or LDAC in terms of median OS, CR/CRi rates and PFS [52]. 

The intensive FLAG-IDA regimen (fludarabine, cytarabine, idarubicin and GCSF) in combination with venetoclax proved feasible and showed high response rates in both newly diagnosed and relapsed or refractory AML patients (median age of 46 years) [44]. CR was achieved in 90% and 61% of newly diagnosed and relapsed or refractory AML patients, respectively. 

## 3. Toxicity of Venetoclax-Based Therapy as Compared to Chemotherapy in AML Patients

As compared to conventional therapies such as azacitidine, LDAC and intensive chemotherapy the treatment with venetoclax-based combination therapy increased the number of adverse events (AE) ≥ grade 3, such as (febrile) neutropenia, anemia and pneumonia [14,31,38,42,52]. Additionally, patients treated with venetoclax-based therapy experienced more common AEs such as nausea, vomiting and diarrhea (Table 2). Increasing the dosage of venetoclax from 400 to 800 mg did not affect the number of patients experiencing AEs ≥ grade 3 [31]. In patients treated with venetoclax monotherapy [52], common AEs were observed more frequently compared to other treatment regimens (Table 2), which can be explained by the fact that venetoclax monotherapy [52] was administered to relapsed or refractory patients, and the other studies treated newly diagnosed patients [14,31,38,42]. In conclusion, venetoclax-based therapy increases the risk of AEs (≥grade 3) and common AEs compared to conventional therapies in AML patients, but there is no increase in AEs upon increasing venetoclax dosage.

## 4. Sensitivity to Venetoclax-Based Therapies in Molecular Defined Subgroups of AML

FLT3-ITD, RUNX1, ASXl1 and TP53 mutations, or a monosomal karyotype have been classified as AML with adverse risk, while NPM1 or biallelic CEBPA mutations are classified as favorable risk, based on treatment with standard induction chemotherapy [4]. AML molecular subtypes showed also enhanced or decreased sensitivity to venetoclax-based therapy. In addition to these molecular aberrancies, epigenetic leukemic cell characteristics may be associated with an enhanced or decreased sensitivity to both chemotherapy and/or venetoclax in AML patients.

## 5. NPM1-, IDH1/2-, TET2- and Relapsed or Refractory RUNX1-Mutated AML Patients Showed Enhanced Sensitivity to Venetoclax

AML patients who received venetoclax-based therapy showed increased response rates and OS when harboring NPM1, IDH1/2, TET2 or RUNX1 mutations as compared to the other cases. NPM1 mutations are present in 20–30% of AML cases. In newly diagnosed NPM1-mutated AML patients treated with venetoclax combined with HMA or LDAC CR/CRi, rates up to 91% were observed, which were higher than the overall cohort (Table 3) [31,37,38,39,40,41,54]. The one-year OS rate of NPM1-mutated cases treated with venetoclax-HMA/LDAC therapy was 80% (median age of 71 years), which was better than treatment of these patients with HMA monotherapy (1-year OS of 12%) or induction chemotherapy (1-year OS of 56%) [56]. The combination of a NPM1 mutation with an FLT3-ITD mutation with a high ratio correlated with reduced sensitivity to treatment with venetoclax-based therapy compared to a sole NPM1 mutation or an NPM1 mutation with a low FLT3-ITD ratio [37].

The mechanism behind the increased sensitivity to venetoclax in NPM1-mutated AML patients remains unclear. NPM1 functions in ribosome biogenesis, genomic stability, p53-dependent stress response and activation of growth-suppressive pathways [57,58,59]. Upon apoptotic stimuli, NPM1 is translocated to the cytoplasm where it inhibits activated caspase-6 and -8, thereby reducing caspase-induced apoptosis [60,61]. As NPM1 mutations always involve a nuclear export signal, cytoplasmic levels of NPM1 are abnormally high in mutated cases, which leads to inhibition of apoptosis. Mutated NPM1 also drives the overexpression of homeobox (HOX) genes, which induce a stem-cell-like state [62,63]. Furthermore, mutated NPM1 is closely associated with FOXM1 inactivation, which is a protein involved in cancer development and chemoresistance [64]. While mutated NPM1 inhibits the apoptotic pathway by inactivating caspases, venetoclax stimulates apoptosis. Possibly, venetoclax sensitivity arises through NPM1-mutation-driven HOX gene expression, since HOX gene expression was shown to be associated with high sensitivity to BCL-2 inhibition in AML cells [65]. In acute lymphocytic leukemia, overexpression of HOX genes is associated with DOT1L-mediated H3K79 hypermethylation, resulting in overexpression of BCL-2 and sensitization to venetoclax [66,67]. Possibly, HOX gene-induced DOT1L-mediated H3K79 hypermethylation is also responsible for the increased sensitivity to venetoclax in NPM1-mutated AML patients. 

Isocitrate dehydrogenases 1/2 (IDH1/2) mutations are observed in 20% of AML patients. IDH produces NADPH from NADP+ by catalyzing the oxidative decarboxylation of isocitrate to alpha-ketoglutarate (αKG) [68]. Mutated IDH acquires a neomorphic function that catalyzes the reduction of αKG to the oncometabolite R-2-hydroxyglutarate (2HG) [69], promoting tumorigenesis through inhibition of ten-eleven translocation (TET) enzymes [70,71]. This inhibition results in aberrant histone methylation and gene expression, which causes impaired differentiation, cell proliferation and DNA damage [70,71,72,73]. AML patients with IDH1/2 mutations treated with venetoclax combined with either HMA or LDAC showed higher CR/CRi rates (56–89%) than the overall patient cohort (50–71%) (Table 3) [31,37,38,39,40,41,48,50,52,54,74]. IDH1/2 mutated AML patients appeared to have better responses with venetoclax-HMA/LDAC than with induction chemotherapy (CR/CRi: 61%) [75]. Responses to this venetoclax-based therapy were rather durable in IDH2 mutated patients, while for patients with IDH1 mutation, this association with prolonged OS was less clear [37]. In addition, in relapsed or refractory IDH1/2-mutated AML patients, venetoclax-based therapy showed increased response rates (25–60%) compared to the overall cohort (12–46%) (Table 3) [39,48,50,54]. Even when treated with venetoclax as monotherapy, CR/CRi rates for IDH2-mutated relapsed and refractory AML patients increased to 33% compared to 19% in patients with wild-type IDH2 [52]. 

Since IDH mutations result in inhibition of TET enzymes, the response of venetoclax-based therapy was also studied in TET2-mutated AML patients [37,48,70]. The presence of a TET2 mutation, similar to the presence of an IDH mutation, drastically increased sensitivity to venetoclax-based therapies in newly diagnosed, relapsed and refractory AML patients, with CR/CRi rates up to 86% compared to 39% in patients with wild-type TET2 (Table 3) [37,48]. The superior response rates to venetoclax-based therapies in IDH1/2- and TET2-mutated AML patients are thought to be caused by the dependency of these molecular subgroups on BCL-2. BCL-2 was identified as being synthetically lethal with IDH mutations in AML, thereby sensitizing IDH-mutated AML cells for venetoclax [76]. This effect may be caused by a decrease in cytochrome C oxidase activity, induced by enhanced 2HG in the IDH-mutated AML cells, which lowers the threshold for venetoclax-induced apoptosis [76]. The induction of HOX genes by the IDH mutation is also associated with increased sensitivity to venetoclax [65,70]. 

Mutations in RUNX1 often lead to loss of function of the RUNX1 protein and create HSCs with aberrant differentiation abilities and resistance to genomic stress [77]. Approximately 10% of AML patients have a mutation in RUNX1 [78], and RUNX1-mutated AML is classified as adverse risk [4]. For newly diagnosed RUNX1-mutated AML patients, venetoclax-based therapy is less effective than for relapsed or refractory RUNX1-mutated AML patients. Similar to the response to induction chemotherapy, RUNX1-mutated newly diagnosed AML patients showed lower CR/CRi rates (50%) to venetoclax-based therapy than patients with wild-type RUNX1 (64%) [37]. In newly diagnosed AML patients who were refractory to venetoclax-based therapy, 40% harbored a RUNX1 mutation, suggesting that the RUNX1 mutation caused reduced sensitivity to venetoclax [37]. However, in most studies investigating relapsed and refractory AML patients after chemotherapy, venetoclax-based treatment resulted in impressive response rates in RUNX1-mutated AML patients (CR/CRi: 35–75%) (Table 3) [39,48,50,54]. Furthermore, in these RUNX1-mutated patients, the median OS was significantly increased (not reached, follow-up of 6.9 months) compared to RUNX1-wild-type patients (5.8 months) [54]. Mutations in RUNX1 resulted in decreased ribosome biogenesis, metabolism and sensitivity for induction of apoptosis in HSCs, creating resistance to endogenous and genotoxic stress [77]. As venetoclax decreases OXPHOS [25,29], inhibition of energy metabolism by both the RUNX1 mutation and treatment with venetoclax may result in enhanced apoptosis, and this might explain the fact that RUNX1-mutated AML patients with an adverse risk and poor response to standard treatment have increased response rates to venetoclax-based therapy [4]. 

## 6. AML Patients with Mutations in FLT3, TP53 and RAS Showed Reduced Sensitivity to Venetoclax

AML harboring FMS-like tyrosine kinase 3 (FLT3) and TP53 mutations are classified as an adverse risk group [4]. FLT3 is a transmembrane tyrosine kinase receptor that plays an important role in normal hematopoietic development, and FLT3 mutations are found in approximately 30% of AML patients [79]. Upon ligand binding, the FLT3 receptor dimerizes and activates downstream signaling pathways regulating transcription, proliferation and apoptosis [80]. FLT3 activation allows for cell survival through the stimulation of BCL-2 and inactivation of the pro-apoptotic protein BAX [81]. In AML, most often, these FLT3 mutations are internal tandem duplications (ITD) or missense mutations in the tyrosine kinase domain (TKD), resulting in constitutive activation of the FLT3 kinase and activation of the RAS/MAPK pathway [82,83,84,85]. 

Co-occurrence of FLT3-ITD mutations with ELN favorable risk mutations, such as mutations in NPM1, generally decreases treatment outcomes [37]. Response rates in FLT3-mutated AML patients treated with intensive chemotherapy are 51–54% with a median OS of 5.8 to 26 months [86,87], while treatment with venetoclax-based therapy resulted in CR/CRi rates ranging from 33% to 72% (Table 3) [31,37,38,39,40,41,48,54,88]. Median OS was 12.5 months for FLT3-mutated patients treated with azacitidine and venetoclax compared to 8.6 when treated with azacitidine alone [89]. 

Relapse after venetoclax-based therapy was associated with clonal expansion of AML cells containing the FLT3-ITD mutation, suggesting that FLT3-ITD-positive cells had a survival advantage during venetoclax treatment [37]. Resistance to venetoclax-based therapy in FLT3-mutated AML patients may arise through multiple mechanisms. Constitutive activation of FLT3, due to the mutation, results in inactivation of BAX, thereby inhibiting apoptosis (Figure 1B) [81]. Moreover, FLT3-ITD AML cells are associated with enhanced expression of MCL-1 [90], creating more dependency on MCL-1 than on BCL-2 for survival (Figure 1B). Additional therapies targeting FLT3 or MCL-1 should be evaluated to improve response and survival for FLT3-mutated AML patients. 

**Table 3 cancers-14-03456-t003:** Response rates for venetoclax-based therapy for different characteristics.

					Total Response	Mutated Response
Mutation	Study	Stage	Combination Therapy	Incidence (%)	CR/CRi (%)	CR/CRi (%)
NPM1	DiNardo (2019) [31]	ND	HMA	16	67	91
	DiNardo (2020) [37]	ND	HMA/LDAC	20	64	72
	DiNardo (2020) [38]	ND	HMA	17	66	67
	Morsia (2020) [39]	ND	HMA	14	50	60
	Pollyea (2020) [40]	ND	HMA	17	71	79
	Wei (2019) [41]	ND	LDAC	13	54	89
	Wang (2020) [54]	R/R	HMA/LDAC	8	23	67
IDH1/2	DiNardo (2019) [31]	ND	HMA	24	67	71
	DiNardo (2020) [37]	ND	HMA/LDAC	28	64	89
	DiNardo (2020) [38]	ND	HMA	25	66	75
	Morsia (2020) [39]	ND	HMA	21	50	56
	Pollyea (2020) [40]	ND	HMA	26	71	86
	Pollyea (2022) [74]	ND	HMA	26	63	79
	Wei (2019) [41]	ND	LDAC	25	54	72
	Aldoss (2019) [48]	R/R	HMA	17	46	60
	DiNardo (2018) [50]	R/R	HMA/LDAC	26	12	27
	Konopleva (2016) [52]	R/R	MONO	12	19	33
	Morsia (2020) [39]	R/R	HMA	12	33	60
	Wang (2020) [54]	R/R	HMA/LDAC	11	23	25
TET2	DiNardo (2020) [37]	ND	HMA/LDAC	23	64	58
	Aldoss (2019) [48]	R/R	HMA	8	46	86
RUNX1	DiNardo (2020) [37]	ND	HMA	23	64	50
	Aldoss (2019) [48]	R/R	HMA	22	46	35
	DiNardo (2018) [50]	R/R	HMA/LDAC	19	12	50
	Morsia (2020) [39]	R/R	HMA	10	33	75
	Wang (2020) [54]	R/R	HMA/LDAC	28	23	36
FLT3	DiNardo (2019) [31]	ND	HMA	12	67	72
	DiNardo (2020) [37]	ND	HMA/LDAC	7	64	33
	DiNardo (2020) [38]	ND	HMA	14	66	72
	Morsia (2020) [39]	ND	HMA	23	50	40
	Pollyea (2020) [40]	ND	HMA	14	71	58
	Wei (2019) [41]	ND	LDAC	23	54	44
	Aldoss (2019) [48]	R/R	HMA	27	46	42
	Aldoss (2020) [88]	ND/R/R	HMA	100	60	60
	Morsia (2020) [39]	R/R	HMA	10	33	50
	Wang (2020) [54]	R/R	HMA/LDAC	8	23	33
TP53	DiNardo (2019) [31]	ND	HMA	25	67	47
	DiNardo (2020) [37]	ND	HMA/LDAC	23	64	44
	DiNardo (2020) [38]	ND	HMA	23	66	55
	Morsia (2020) [39]	ND	HMA	21	50	44
	Pollyea (2020) [40]	ND	HMA	20	71	53
	Wei (2019) [41]	ND	LDAC	14	54	30
	Aldoss (2019) [91]	R/R	HMA	14	46	46
	Morsia (2020) [39]	R/R	HMA	29	33	40
	Wang (2020) [54]	R/R	HMA/LDAC	8	23	0
K/NRAS	DiNardo (2020) [37]	ND	HMA	19	64	33
	Aldoss (2019) [48]	R/R	HMA	16	46	36
	Wang (2020) [54]	R/R	HMA/LDAC	8	23	0
Other						
Prior HMA	DiNardo (2020) [37]	ND	HMA	6	64	20
	Wei (2019) [41]	ND	LDAC	29	54	33
	Winters (2019) [43]	ND	HMA	12	58	0
	Goldberg (2017) [92]	R/R	HMA/LDAC	76	24	25
	Morsia (2020) [39]	R/R	HMA	36	33	40

Abbreviations: CR/CRi, complete remission/complete remission with incomplete blood count recovery; HMA, hypomethylating agents; LDAC, low dose cytarabine; mOS, median overall survival; ND, newly diagnosed; R/R, relapsed/refractory.

AML patients with RAS or PTPN11 mutations, although not classified in the ELN classification system [4], showed decreased OS when treated with induction chemotherapy, with a median OS of 12 months for RAS-mutated AML patients and compared to 30.1 months for the other patients [93], and a median OS of 13.4 months for PTPN11-mutated AML patients compared to 19.2 months for wild-type PTPN11 AML patients [94]. In addition, when treated with venetoclax-based therapy, the presence of RAS mutations in newly diagnosed, relapsed and refractory AML patients is associated with poor response rates with CR/CRi rates of 0–36% compared to 23–64% for the total patient cohort, and a shorter median OS (3.8 months) compared to the patients with wild-type NRAS (7.4 months) (Table 3) [37,48,54]. In AML patients that progress or relapse after treatment with venetoclax and azacitidine, 27% of the patients had a N/KRAS mutation, and 22% a PTPN11 mutation [32,95], suggesting that these mutations are more frequently present in refractory or relapsed disease. The reduced sensitivity to venetoclax in RAS- and PTPN11-mutated AML patients can be explained by the activation of the RAS/MAPK signaling pathway [96], allowing the AML cells in RAS- and PTPN11-mutated cases to use multiple sources of energy for cell survival, including fatty acid and amino acid metabolism, glycolysis and upregulation of OXPHOS [97,98]. Moreover, AML with a PTPN11 mutation often has a more monocytic cell type [99], which generally has increased dependency on MCL-1 instead of BCL-2 [100], thereby decreasing the efficacy of venetoclax. The poor response rates of RAS and PTPN11-mutated AML patients to venetoclax-based therapies suggest that for these patients, conventional chemotherapy is the best treatment option and in order to increase response rates other additional therapy, MCL-1 or MEK1/2 inhibitors [101,102] could be a future treatment strategy. 

Since venetoclax inhibits BCL-2, a key member of the apoptosis pathway, additional mutations or epigenetic and transcriptional changes affecting apoptosis may influence treatment response. TP53 is one of the most frequently mutated genes in all types of human cancer (reviewed in [103,104] and plays an important role in inducing apoptosis. Mutations in TP53 lead to uncontrolled proliferation and cancer formation associated with poor prognosis [103,104,105]. While TP53 mutations play an important role in human solid tumors, their role in AML is less prominent. In the TCGA AML cohort, 8% of TP53 mutations are found in newly diagnosed AML patients [78]. In these patients, the response to conventional chemotherapy is poor (2-year OS of 12.8–14%), relapse often occurs [106,107,108], and CR/CRi rates are decreased in TP53-mutated patients compared to the overall cohort treated with venetoclax-based therapy (0–55% TP53-mutated and 23–71% in wild-type TP53 patients) (Table 3) [31,37,38,39,40,41,54,91]. The presence of a mutation in TP53 also predicts a poor response and OS after treatment with venetoclax-based therapy [31,39], with a median OS of 1.9 months in TP53-mutated AML patients compared to 7.4 months in patients with wild-type TP53 [54]. Furthermore, 32% of the relapsed AML patients had an expansion of TP53-mutated cells after venetoclax-based treatment, suggesting that the presence of mutated-TP53 reduced sensitivity to venetoclax or enhanced relapse-initiating potential [37]. In AML cells, TP53 and one of its transcriptional targets BAX showed to be key regulators of venetoclax sensitivity [109]. Knockout of TP53 resulted in decreased MCL-1 and BCL-2 levels, ultimately decreasing sensitivity to venetoclax [110], suggesting that TP53-mutated AML cells are more dependent on MCL-1 than BCL-2 for survival. Additionally, alterations in expression of p53’s transcriptional targets affect mitochondrial homeostasis and cellular metabolism, causing increased oxygen consumption and mitochondrial electron transport [109]. Simultaneous BCL-2 inhibition and p53 activation was shown to be synthetically lethal in AML cells, addressing the importance of the presence of a mutation in TP53 when applying venetoclax-based therapy [111]. 

## 7. Venetoclax Sensitivity Is Decreased in More Mature AML Cells

AML patients with a more differentiated monocytic AML were significantly more refractory to venetoclax-based treatment than AML with more immature disease [100]. The monocytic AML type is characterized by loss of the primitive CD117 marker and upregulation of monocytic markers CD11b, CD68 and CD64 on the AML cells, and is classified as FAB-M5 [100]. Of the M5 AML cases, 62% were refractory to venetoclax and azacitidine, whereas only 0–8% of the less differentiated cell types (non-M5) were [100]. The median OS of patients with a monocytic AML was 3.0 months compared to 17.3 months for the non-M5 AML subtypes. FAB-M5 AML harbored significantly more FLT3-ITD mutations than non-M5 AML subtypes. The resistance of monocytic AML cells to venetoclax and azacitidine can be explained by the significantly increased presence of FLT3-ITD mutations [112] and the loss of BCL-2 [100]. Similarly, during normal monocytic development, upon monocytic maturation of AML cells, a decrease in BCL-2 gene expression was observed [27,113]. Furthermore, monocytic AML cells showed increased energy metabolism through upregulation of OXPHOS [100]. MCL-1 is a key mediator of OXPHOS and is essential for viability of monocytic AML cells [100,110]. Similarly, in FLT3-mutated AML, the monocytic AML cells demonstrated an increased dependency on MCL-1 instead of BCL-2, and treatment of monocytic AML with an MCL-1 inhibitor together with azacitidine significantly reduced cell survival compared to treatment with venetoclax [114]. The identification of a biomarker reflecting the differentiation stage of the AML may add to the prediction of sensitivity to venetoclax. More differentiated or monocytic AML might be treated with a combination of azacitidine and an MCL-1 inhibitor or an FLT3 inhibitor instead of venetoclax. 

## 8. Prior Treatment with Hypomethylating Agents Decreased Effectiveness of Venetoclax

AML patients that received prior HMA therapy respond poorly to venetoclax-based therapy, in both secondary AML and relapsed or refractory AML. Secondary AML patients that received prior HMA therapy, for instance as treatment for myelodysplastic syndrome, achieved CR/CRi rates of 0–33% when treated with venetoclax-based therapy (Table 3) [37,41,43]. Likewise, the median OS in these patients treated was only 3.7 months compared to 15.5 months in patients with no prior HMA treatment [37]. Similarly, in AML patients that relapsed or were refractory after HMA treatment, response rates to venetoclax-based therapy were poor (CR/CRi: 25–40%) (Table 3) [39,92]. The poor response rates and resistance to venetoclax combined with HMAs is possibly driven by the emergence or expansion of HMA resistant subclones and/or HMA-induced mutations that drive resistance to venetoclax [115,116,117,118]. 

## 9. Mechanisms Determining Sensitivity of AML to Venetoclax

Resistance to venetoclax arises through various mechanisms. Venetoclax prevents BCL-2 from binding the pro-apoptotic proteins BAX and BIM, which increases permeability of the mitochondrial outer membrane, thereby releasing cytochrome C and inducing apoptosis (Figure 1A) [17,18]. In some AML patients, leukemic cell survival depends on other functional anti-apoptotic proteins than BCL-2, for example MCL-1 (Figure 1B) [119]. These patients may have primary resistance to venetoclax-based therapies due to the upregulation of MCL-1, which is stabilized through epigenetic modifications caused by the activation of the RAS/MAPK signaling pathway [96,111]. As previously mentioned, AML cells with a more monocytic AML cell type or the FLT3, RAS, PTPN11 or TP53 mutations have increased dependency on MCL-1 instead of on BCL-2 [90,100,101,110,120]. Treatment with venetoclax of these AML cases will not prevent MCL-1 from inhibiting the activation of BAX and BIM, blocking the activation of the apoptosis pathway (Figure 1B). Therefore, BCL-2 inhibition by venetoclax will not be sufficient for inducing cell death, and in the future, a combination therapy that includes the inhibition of MCL-1 could be considered. 

Furthermore, resistance to venetoclax may occur when AML cells can increase OXPHOS or use other resources than OXPHOS for energy. Venetoclax decreases OXPHOS, inhibiting amino acid metabolism and inhibiting cell proliferation, which specifically targets LSCs [25] (Figure 1A). In contrast to HSCs, LSCs depend primarily on amino acids and OXPHOS for their mitochondrial metabolism [30]. Resistance to venetoclax can arise when OXPHOS is increased or when other energy sources, such as glycolysis and phospholipids, are used to compensate for low cellular OXPHOS and amino acid levels (Figure 1B) [97,98]. As said, monocytic AML cells are able to increase OXPHOS, and RAS- and PTPN11-mutated AML cells are able to use fatty acid metabolism and glycolysis to compensate for low OXPHOS and low amino acid levels, thereby reducing sensitivity to venetoclax [97,98,100]. In relapsed or refractory AML patients, an elevation in nicotinamide metabolism was observed after venetoclax-based therapy [114]. Nicotinamide is an import component of the oxidation process, and high levels of nicotinamide result in amino acid metabolism and fatty acid oxidation [114], suggesting that AML cells could use alternative energy sources by stimulation of nicotinamide metabolism. These AML cells are expected to be more resistant to venetoclax than cells that only use OXPHOS for energy. 

BCL-2 and other members of the apoptotic pathway need to be present and need to function properly for venetoclax to be active (Figure 1B). As mentioned, monocytic and TP53-mutated AML cells have decreased levels of BCL-2, causing decreased sensitivity to venetoclax [100,110]. In addition, inactivation of BAX through FTL3 mutations prevents the induction of apoptosis by venetoclax [81]. In CLL patients, mutations in BCL-2 are found that can prevent or reduce the binding of venetoclax to the BCL-2 protein, thereby inhibiting its efficacy [121,122,123]. These BCL-2 mutations are usually acquired when CLL patients relapse. Mutations in BAX, inhibiting its localization to the mitochondrial outer membrane and thereby inhibiting the release of cytochrome C, are also found in CLL and prevent venetoclax-induced apoptosis [124]. In AML patients who relapse after treatment with venetoclax, mutations in BCL-2 and BAX play a less prominent role in the development of resistance [125]. However, the relevance of BCL-2 and BAX mutations in driving venetoclax resistance in AML patients still has to be elucidated in more detail. 

## 10. Prediction of Active Venetoclax-Based Combination Therapies for AML

Research has been focusing on possible combination therapies with venetoclax, specifically targeting tumor characteristics, hopefully reducing both primary and acquired resistance. Some of these combinations are already tested in clinical settings, while others have only shown promising results in AML cell lines thus far. 

IDH1/2- mutated AML patients have higher response rates to venetoclax-based treatment than to IDH inhibitor-based treatment [126,127,128]. Therefore, the standard of care for IDH-mutated patients ineligible for standard chemotherapy remains as venetoclax with HMA [129]. However, IDH inhibitors have a favorable toxicity profile in AML therapy (reviewed in [130]). Therefore, combining venetoclax with IDH inhibitors might minimize adverse events with similar response rates. A phase 1b/2 study testing multiple combinations of ivosidenib plus venetoclax in IDH1-mutated AML and myelodysplastic syndrome found CR/CRi rates of 67% to 85% with common adverse events (≥grade 3), including febrile neutropenia (28%) and pneumonia (24%) [131]. Recently, more clinical trials started investigating the addition of IDH inhibitors, such as enasidenib and ivosidenib, to venetoclax-based therapies for IDH-mutated AML patients (NCT03471260, NCT04092179, NCT04774393). 

Recent studies are focusing on the use of FLT3 inhibitors such as midostaurin and gliteritinib in the treatment of FLT3-mutated AML patients (reviewed in [132]). The addition of midostaurin to intensive chemotherapy improved the 4-year OS from 44% to 51% [87], and gilteritinib monotherapy improved median OS to 9.3 months compared to salvage chemotherapy (median OS 5.6 months) for FLT3-mutated AML patients [133]. Triple therapy combining venetoclax, decitabine and an FLT3 inhibitor achieved CR/CRi in 92% in a small cohort of newly diagnosed AML patients [134] compared to the aforementioned 33–72% response rates with the combination of venetoclax and decitabine (Table 3) [31,37,38,39,40,41]. In relapsed or refractory AML patients, 62% achieved CR/CRi with this combination [134], compared to 33–60% in studies using dual therapy (Table 3) [39,48,54,88]. The median 2-year OS for newly diagnosed and relapsed or refractory AML patients, when treated with triple therapy, was 80% and 29%, respectively [134]. In addition, in a retrospective analysis in FLT3-mutated patients, CR/CRi rates improved to 93% with triple therapy (low intensity chemotherapy, FLT3 inhibitor and venetoclax) compared to 70% with double therapy (low intensity chemotherapy and an FLT3 inhibitor) [135]. Possibly, future treatment options for FLT3-mutated AML patients should include the combination of venetoclax together with an FLT3 inhibitor. 

An effective combination strategy for AML patients that acquired more dependency on MCL-1 for survival than on BCL-2 is the combination of venetoclax with an MCL-1 inhibitor [111,120]. Mainly, monocytic and FLT3-, RAS-, PTPN11- and TP53-mutated AML cells showed resistance to venetoclax caused by higher levels of MCL-1, suggesting that these subgroups of AML cases could be efficiently treated with the combination of venetoclax and a MCL-1 inhibitor [97,100]. Simultaneous inhibition of BCL-2 and MCL-1 synergistically enhances apoptosis in AML cells [136,137]. Even in AML cells that weakly express MCL-1, MCL-1 inhibitors synergized with venetoclax [138], indicating that the addition of MCL-1 inhibitors to venetoclax treatment might be beneficial for all AML patients. Ongoing phase I clinical trials are testing different MCL-1 inhibitor-based therapies for AML patients, including the combination with venetoclax (NCT03672695, NCT02979366, NCT04629443).

The increased dependency on MCL-1 is caused by activation of the MAPK signaling pathway, which stabilizes the MCL-1 protein and prevents its degradation [96,111]. Inhibition of the MAPK signaling pathway with MEK1/2 inhibitors synergizes with venetoclax to prevent survival in AML cells [102]. Simultaneous inhibition downregulated MCL-1 levels and disrupted the binding of BIM to both BCL-2 and MCL-1 and released BIM to initiate apoptosis [102]. Venetoclax combined with MEK1/2 inhibitors might therefore improve response rates in RAS-, PTPN11- and FLT3-ITD-mutated AML patients since these mutations activate the MAPK signaling pathway [83,96]. 

Since OXPHOS plays an important role in the activity of venetoclax, research is focusing on inhibition of the mitochondrial complexes, by for example tedizolid, metformin and AXL inhibitors, thereby potentially increasing sensitivity of AML cells to venetoclax [139,140,141]. For instance, tedizolid reduces the expression of a cytochrome C subunit and combined with venetoclax effectively reduces AML cell viability, compared to either treatment alone [142]. Metformin decreases electron transport chain complex I activity, altering mitochondrial metabolism [143]. Combined with venetoclax, metformin induces a synergistic pro-apoptotic effect on AML cells by reducing MCL-1 expression [144]. AXL is a growth-arrest specific gene that is increased in AML cells, specifically stem cells [145]. Alone and in combination with venetoclax, AXL inhibition inhibits mitochondrial metabolism and decreases OXPHOS [141]. Relapsed or refractory AML cells showed increased dependency for survival on fatty acid metabolism [30], and it was shown that simultaneous inhibition of BCL-2 and fatty acid uptake by a CD36 inhibitor decreased OXPHOS and viability in relapsed AML cells [30,98]. 

Lastly, ongoing research is investigating novel possible combination therapies to enhance the response to venetoclax in AML cells by targeting specific pathways. Activation of the CXCL12-CXCR4 pathway by the adhesion molecule CD44 increases the level of embryonic stem cell transcription factors causing resistance to venetoclax [146]. Therefore, CD44 is a potential therapeutic target to resensitize AML cells to venetoclax. Inactivation of FOXM1, an important protein in cancer development and chemoresistance [64], as part of a FOXM1-AKT-positive regulation circuit, effectively sensitizes venetoclax-resistant AML cells [147], also making FOXM1 an interesting therapeutic target to combine with targeting BCL-2. 

## 11. Conclusions

The addition of venetoclax to induction chemotherapy may be beneficial for a particular subgroup of AML patients. It seems likely that most suitable patients can be identified by their mutational profile, with high sensitivity to venetoclax in NPM1-, IDH1/2-, TET2- and relapsed or refractory RUNX1-mutated AML patients. AML patients harboring FLT3, TP53, RAS or PTPN11 mutations, monocytic AML, or AML cases pre-treated with HMAs show reduced sensitivity to venetoclax-based therapies. Reduced sensitivity to venetoclax is caused by an increased dependency on other anti-apoptotic proteins, such as MCL-1, the ability to increase OXPHOS or the ability to use alternative sources for energy metabolism than OXPHOS, and aberrant levels or mutations in BCL-2 and BAX. Future research is investigating new ways to overcome the reduced sensitivity to venetoclax of AML cells with novel combination therapies such as MCL-1, FLT3, MEK1/2 and mitochondrial complex inhibitors, which need to be evaluated in a clinical setting.

## Figures and Tables

**Figure 1 cancers-14-03456-f001:**
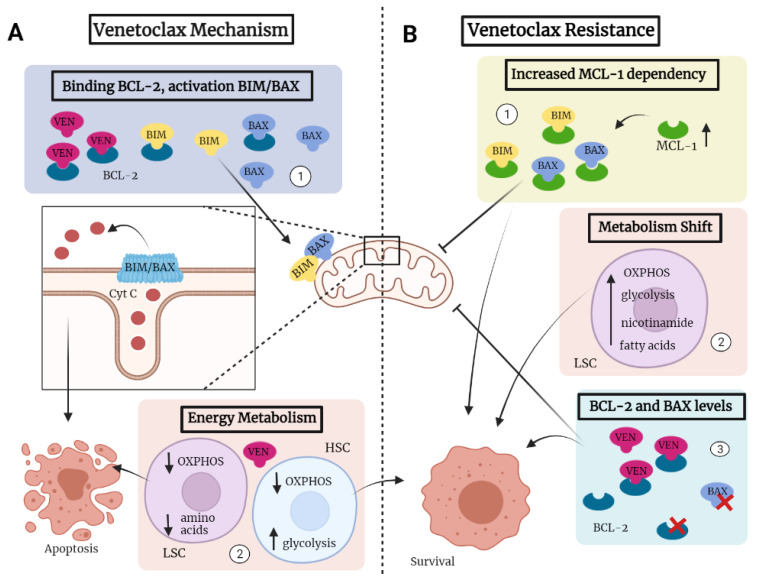
Mechanism of action and resistance in venetoclax treatment. (**A**) Mechanism of action: 1. Venetoclax prevents the inhibitory interaction between BCL-2 and the pro-apoptotic proteins BAX and BIM. BAX and BIM increase mitochondrial outer membrane permeability, releasing Cyt C and initiating apoptosis. 2. Venetoclax reduces OXPHOS and amino acid uptake in LSC, normal HSC compensate through increased glycolysis. (**B**) Mechanism of venetoclax resistance: 1. Increased MCL-1 dependency prevents mitochondrial localization of BIM and BAX, 2. Other energy sources than OXPHOS are used for energy metabolism. 3. A decrease in or mutations in BCL-2 prevent venetoclax from binding, and mutations in BAX inhibit mitochondrial localization. Abbreviations: Cyt C; cytochrome C, HSC; hematopoietic stem cell, LSC; leukemic stem cell, OXPHOS; oxidative phosphorylation, VEN; venetoclax.

**Table 1 cancers-14-03456-t001:** Overview of studies with venetoclax-based therapy for AML patients.

				Risk Stratification (%)					Overall Outcome		CR/CRi Outcome (Months)	
Study	Stage	*n*	Median Age (Range)	Favorable	Intermediate	Adverse	NA	Combination Therapy	mOS (Months)	CR/CRi (%)	mOS	PFS
Asghari (2019) [36]	ND	41	75 (47–86)	24.4		58.5	17.1	HMA	13.8	56	NA	5.2
DiNardo (2019) [31]	ND	145	75 (65–86)	-	51	49	-	HMA	17.5	67	NA	11.3
DiNardo (2020) [37]	ND	58	74 (62–87)	-	62.1	34.5	3.4	HMA	15.1	69	NA	22.4
		23	73 (66–78)	4.3	60.9	26.1	8.7	LDAC	10.7	52	NA	17.1
DiNardo (2020) [38]	ND	286	76 (49–91)	-	64	36	-	AZA	14.7 (11.9–18.7)	37	NA	NA
DiNardo (2021) [44]	ND	29	45 (20–65)	17.2	44.8	37.9	-	FLAG-IDA	NR	90	NA	NR
Morsia (2020) [39]	ND	44	65 (18–79)	7.1	26.2	66.7	-	HMA	11 (8–23)	50	17 (9-NR)	NR (11-NR)
Pollyea (2020) [40]	ND	84	75 (61–90)	-	60	39	1	AZA	16.4 (11.3–24.5)	71	NA	21.9 (15.1–30.2)
		31	72 (65–86)	-	52	48	-	DEC	16.2 (9.1–27.8)	74	NA	15.0 (7.2–30.0)
Wei (2019) [41]	ND	82	74 (63–90)	-	60	32	8	LDAC	10.1 (5.7–14.2)	54	18.4 (14.0-NR)	8.1 (5.3–14.9)
Wei (2020) [42]	ND	143	76 (36–93)	1	63	33	3	LDAC	8.4 (5.9–10.1)	34	NA	4.7 (3.7–6.4)
Winters (2019) [43]	ND	33	72 (33–85)	-	-	67	33	AZA	12.7 (5.8-NR)	63	NA	10.7 (4.5-NR)
Aldoss (2018) [47]	R/R	33	62 (19–81)	9.1	33.3	54.5	3	HMA	NR	51	NR	8.9 (3.2–10.6)
Aldoss (2019) [48]	R/R	90	59 (18–81)	8	26	66	-	HMA	7.8 (5.9–15.5)	46	16.6 (13.5–26.8)	8.9 (5.9–15.2)
Asghari (2019) [36]	R/R	31	63 (25–77)	25.8		45.2	29.0	HMA	4.9	28	NR	5.2
Byrne (2020) [49]	R/R	21	65 (35–74)	4.8	47.6	47.6	-	HMA	7.8 (0.2–12.1)	38	NR	NA
DiNardo (2018) [50]	R/R	43	68 (25–83)	-	53	47	-	HMA/LDAC	3.0 (0.5–8.0)	12	4.8	NA
DiNardo (2021) [44]	R/R	23	47 (22–66)	26	13	61	-	FLAG-IDA	NR	61	NA	11 (2-NR)
Gaut (2020) [51]	R/R	14	58 (41–79)	-	28.6	71.4	-	HMA/LDAC	4.7 (NA)	21	NR	NR
Konopleva (2016) [52]	R/R	32	71 (19–84)	-	-	-	100		4.7 (2.3–6.0)	19	NA	2.3 (1.0–2.7)
Morsia (2020) [39]	R/R	42	65 (18–79)	7.1	26.2	66.7	-	HMA	5 (3–9)	33	15 (5-NR)	8 (1–20)
Ram (2019) [53]	R/R	23	76 (41–92)	9	48	43	-	HMA/LDAC	5.6 (4.9–6.2)	43	10.8 (6.2–15.4)	NA
Wang (2020) [54]	R/R	40	63 (20–88)	12.5	17.5	70	-	HMA/LDAC	6.6 (0.7–16.3)	23	NR	NA

Abbreviations: AZA, azacitidine; CR/CRi, complete remission/complete remission with incomplete blood count recovery; DEC, decitabine; FLAG-IDA, fludarabine, cytarabine, granulocyte colony-stimulating factor and idarubicin; HMA, hypomethylating agents.

**Table 2 cancers-14-03456-t002:** Adverse events > grade 3 and common adverse events for AML therapies.

		Adverse Events ≥ Grade 3 (%)				Common Adverse Events (%)		
Study	Therapy	Febrile Neutropenia	Neutropenia	Anemia	Pneumonia	Nausea	Vomiting	Diarrhea
Dombret (2015) [14]	AZA	28	36	16	19	27	14	12
	LDAC	30	25	23	19	22	11	5
	IC	31	33	14	5	43	7	21
DiNardo (2019) [31]	AZA + VEN (400 mg)	38	NA	31	NA	62	31	52
	AZA + VEN (800 mg)	35	NA	24	NA	62	27	49
DiNardo (2020) [38]	AZA + placebo	19	28	20	25	35	23	33
	AZA + VEN (400 mg)	42	42	26	56	44	30	41
Wei (2020) [42]	LDAC + placebo	29	16	22	16	31	13	16
	LDAC + VEN (600 mg)	32	46	25	20	42	25	28
Konopleva (2016) [52]	VEN (800 mg)	31	NA	NA	19	59	41	56

Abbreviations: AZA, azacitidine; IC, induction chemotherapy; LDAC, low dose cytarabine; NA, not available; VEN, venetoclax.

## Data Availability

Not applicable.

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
