# Peer review of "Targeting Acute Myeloid Leukemia with Venetoclax; Biomarkers for Sensitivity and Rationale for Venetoclax-Based Combination Therapies"

_cancers, 2022, doi:10.3390/cancers14143456_

Round 1

Reviewer 1 Report

This is a superb review on venetoclax use in AML, its mechanism of action and resistance pathways.

I do consider this paper suitable to published in Cancers as soon as possible.

Incidentally, "tedizoled" should be changed into "tedizolid"

Reviewer 2 Report

This is an excellent review of Venetoclax in AML.

The study elaborates on the efficacy and safety of the agent compared to conventional chemotherapy for treatment of AML patients. Moreover, it gives a comprehensive overview of AML genetics and responses to Venetoclax. 

The authors discuss venetoclax in monocytic disease.  However, should the authors briefly mention specifically the impact of venetoclax on KMT2Ar leukemias (11q23)? Perhaps they could mention therapy-related AML. 

Venetoclax is now employed in post transplant setting. Should the authors talk about its role in this setting? 

Some of these AML cases can present with extramedullary disease. Should the author discuss about the role of venetoclax in CNS or extramedullary disease? It is nowadays used in r/r setting and readers would like to know more about this.